# Evaluation of Intestinal Microbial Metabolites in Preterm Infants with Different Initial Feeding Methods by In Vitro Fermentation Modeling System

**DOI:** 10.3390/microorganisms10071453

**Published:** 2022-07-19

**Authors:** Yunwei Li, Jingjing Jiang, Liying Zhu, Xin Wang, Weilin Wan, Danhua Wang, Zhenghong Li

**Affiliations:** 1Peadiatric Department, Peking Union Medical College Hospital, Chinese Academy of Medical Science and Peking Union Medical College, Beijing 100730, China; liyunwei@pumch.cn (Y.L.); jiangjingjing@pumch.cn (J.J.); wanweilin-5745@hotmail.com (W.W.); danhuawang10@126.com (D.W.); 2State Key Laboratory for Managing Biotic and Chemical Threats to the Quality and Safety of Agro-Products, Institute of Food Research, Zhejiang Academy of Agricultural Sciences, Hangzhou 310021, China; zhuly@zaas.ac.cn (L.Z.); wangx@zaas.ac.cn (X.W.); 3Peadiatric Department, State Key Laboratory of Complex Severe and Rare Diseases, Peking Union Medical College Hospital, Chinese Academy of Medical Science and Peking Union Medical College, Beijing 100730, China

**Keywords:** human milk, preterm infants, initial feeding methods, intestinal metabolites

## Abstract

We aim to explore the intestinal microbial metabolites in preterm infants with noninvasive methods and analyze the effects of initial feeding methods. Preterm infants with gestational weeks lower than 34 were recruited for fecal sample collection every 7 days. Fecal pH, ammonia, bile acid, and secretory IgA (sIgA) were tested. A 1:10 fecal slurry was inoculated into different culture media containing different carbohydrates as the only carbon source: lactose (LAT), fructooligosaccharide (FOS), galactooligosaccharide (GOS), and 2′-fucosyllactose (FL2). After 24 h of anaerobic culture through an in vitro fermentation system, air pressure difference, carbohydrate degradation rate, and short-chain fatty acids (SCFAs) content in fermentation pots were measured. Preterm infants were assigned into two groups: group A, preterm infants fed by human milk, including mother’s own milk and donor human milk (DHM); group B, preterm infants fed by preterm formula at first 3 days and fed by human milk (including mother’s own milk and DHM) from day 4 to discharge. Group A included 90 samples and group B included 70 samples. Group A had lower fecal pH (*p* = 0.023), ammonia (*p* = 0.001), and bile acids (*p* = 0.025). Group B also had higher fecal sIgA levels, both in OD (*p* = 0.046) and concentration (*p* < 0.0001) methods. Carbohydrates degradation rates in group A were higher than group B, especially in LAT medium (*p* = 0.017) and GOS medium (*p* = 0.005). Gas production amount had no significant difference in all four media. Several different SCFAs in four kinds of different culture media in group A were higher than in group B, but valeric acid was lower in group A. The initial feeding methods may affect the preterm infants’ intestinal microecology and microbial metabolites for at least several weeks.

## 1. Introduction

Due to the immature development of systems in preterm infants, one of the ultimate goals of nutritional support is to help them establish exclusive enteral nutrition (EEN), which is also a key factor for the successful treatment of preterm infants [1]. There have been many studies focusing on influencing factors of feeding intolerance in preterm infants, such as maturity, infection, and feeding methods [2]. However, due to the small size and low blood volume of preterm infants, it is difficult to carry out invasive examinations, so the assessment method of intestinal environment is very limited. At present, most studies only explore the intestinal flora of preterm infants by their feces. Along with the advancement of research, researchers have gradually realized that in addition to direct interaction between gut microbes and human body, many microbes play a key role through various forms of microbial metabolites. The byproducts of human intestinal microorganisms include not only various soluble metabolites, but also different types of gas [3], which can act on the digestive system and even other human systems at the metabolic level. Along with the development of technology, the in vitro anaerobic fermentation system [4], which simulates the intestinal environment, makes it possible to explore intestinal microbial metabolites with non-invasive method.

Human milk is the first choice to feed preterm infants, but most mothers of preterm infants can not express enough milk in the first few days, and their babies must be fed with formula. When mother’s own human milk is not available, pasteurized donor human milk (DHM) is the second choice [5]. At present, worldwide guidelines for feeding preterm infants all advocate that the use of preterm formula milk should only be considered when both mother milk and DHM are not available. From February 2019, we established our human milk bank, so all the preterm infants of less than 1500 g could be fed with human milk, including mother’s own milk and DHM, even in the first few days. In this study, we want to explore the importance of feeding in the first few days in preterm infants and the effect on intestinal microbial metabolites by application of the in vitro anaerobic fermentation system.

## 2. Materials and Methods

### 2.1. Study Population

We conducted a prospective observational study at the Peking Union Medical College Hospital (PUMCH, Beijing, China), which was approved by the Ethics Committee of the Peking Union Medical College Hospital (Date 24 January 2018). All the parents signed informed consent.

Inclusion criteria: (1) preterm infants less than 34 gestational weeks, (2) preterm infants delivered at PUMCH from February 2018 to December 2019, (3) preterm infants hospitalized at neonatal intensive care unit (NICU) of PUMCH, (4) preterm infants who were fed by human milk from day 4 to discharge, (5) parents signed informed consent. 

Exclusion criteria: (1) preterm infants with digestive tract abnormality, nervous system abnormality, congenital immunodeficiency, metabolic disorder, (2) preterm infants were transferred to other hospitals or passed away during hospitalization, (3) changing to formula due to social or pathological factors, (4) demanding withdrawal from research. 

Preterm infants were assigned into two groups according to their enteral feeding: group A, preterm infants fed by human milk, including mother’s own milk and DHM; group B, preterm infants fed by preterm formula for first 3 days and fed by human milk (including mother’s own milk and DHM) from day 4 to discharge.

### 2.2. Data Collection

Clinical data were collected including medical history and recorded feeding methods, feeding amount, body weight, treatment (including antibiotic use, probiotics use), other complications.

### 2.3. Fecal Sample Management

Fecal samples were collected every 7 days until the infants were 34 corrected gestational weeks or discharged. Fecal samples were defected naturally on diapers. Only collectable samples with weight over 1 g were used. Within 6 h after defecation, fecal samples were delivered to lab for immediately following procedure. Fecal pH was tested directly by pH meter (PH5S puncture pen type PH meter, Shanghai Sanxin Instrument, China), and 0.8 g fresh fecal samples were homogenized with 8 mL of 0.1 M anaerobic phosphate-buffered saline (pH 7.0) to make 10% (wt/vvol) fecal slurries by an automatic fecal homogenizer (Halo Biotechnology Co., Ltd., Jiangsu, China). The remaining fresh fecal samples and were preserved at −30 °C for further analysis.

### 2.4. Batch Culture Fermentation

Batch culture fermentation was conducted using the procedure described by Lei et al. [6]. The basic growth medium VI contained the following: 3.0 g/L tryptone; 4.5 g/L yeast extract; 0.05 g/L hemin, 3 g/L peptone; 0.8 g/L L-cysteine hydrochloride; 0.4 g/L KH_2_PO_4_; 2.5 g/L KCl; 4.5 g/L NaCl; 0.45 g/L MgCl_2_·6H_2_O; 0.2 g/L CaCl_2_·6H_2_O; 0.2 g/L MgSO_4_·7H_2_O, 1 g/L resazurin [7]. In order to assess the degradation and utilization of different carbon sources by the human fecal microbiome, 8.0 g/L each of lactose (LAT), fructooligosaccharide (FOS), galactooligosaccharide (GOS), and 2′-fucosyllactose (FL2) were added into the growth medium as the sole carbon source. The medium was adjusted to pH 6.5 before sterilization via autoclave. 5 mL of test medium was dispensed into a 10-mL bottle under anaerobic conditions, which was made by filling nitrogen. 500 μL fresh fecal slurry, immediately after being produced, was added into the prepared batch fermentation bottle in anaerobic growth media. Gas pressure inside the bottle was measured immediately after inoculation. After fermentation at 37 °C for 24 h, gas pressure inside the bottle was measured again. Air pressure difference (kpa) was recorded. The remaining fresh fecal slurry and bottles after fermentation were preserved at −30 °C for further analysis.

### 2.5. Thin-Layer Chromatography

The carbon source degradation products were detected by thin-layer chromatography (TLC) analysis. Briefly, samples (0.2 mL) were loaded onto a pre-coated silica gel-60 TLC aluminum plate (Merck, Darmstadt, Germany). After being developed with a solvent system consisting of formic acid/n-butanol/water (6:4:1, *v*:*v*:*v*), the plate was soaked in orcinol reagent (Sigma-Aldrich Co., LLC., St. Louis, MO, USA) and visualized at 120 °C for 1 min [8]. The scanned TLC profiles were analyzed by Quantity One (BioRad, Hercules, CA, USA) and the amounts of carbon source in each sample were quantified. Degradation rates were calculated as the percentage of total amounts at 0 h subtracted from those at 24 h after fermentation.

### 2.6. Short Chain Fatty Acids Analysis

The concentration of short chain fatty acids (SCFAs) was determined by gas chromatography (GC) [9]. The amounts of acetic acid, propionic acid, butyric acid, isobutyric acid, valeric acid, and isovaleric acid in the culture filtrates were determined by gas chromatography (Shimadzu, GC-2010 Plus, Kyoto City, Japan) equipped with a DB-FFAP column (0.32 mm × 30 m × 0.5 μm) (Agilent Technologies, Santa Clara, CA, USA) using an H_2_ flame ionization detector. Crotonic acid (trans-2-butenoic acid) was used as an internal standard.

### 2.7. Enzyme-Linked Immunosorbent Assay (ELISA)

10% fecal slurry was used to measure total secretory IgA (sIgA). The 96 plates were washed in PBS and blocked in PBS with 1% BSA. Samples were diluted 1/300, and a twofold serial dilution was made. Samples were incubated at 37 °C for 90 min. Total sIgA was quantified by ELISA using a Human Secretory IgA ELISA detection Kit (Elabscience, Wuhan, China). The measurement was taken according to the manufacturer’s instructions. The linear regression equation of the standard curve was calculated by the concentration of the standard substance and the OD value, and the OD value of the sample was substituted into the equation to calculate the sample concentration, which was multiplied by the dilution multiple, namely the actual concentration of the sample.

### 2.8. Correlation Coefficients and Statistical Analysis

Analysis of data was carried out by using IBM SPSS Statistical software (IBM Corp., Armonk, NY, USA). The presence of significant differences among groups was assessed by applying the Mann–Whitney U test for comparing two groups, respectively, while the χ2 test was used for discrete data. In each case, a *p*-value < 0.05 was considered statistically significant.

## 3. Results

A total of 50 preterm infants of gestational age 25–33 weeks were recruited, and 160 fecal samples were collected. Group A included 90 samples and formula milk included 70 samples. According to the clinical situation of premature infants, not all volunteers can defecate within the time range of collecting fecal samples, so there can be leakage of sample detection timing. Even if there is success in the sample collection, the sample weight may not be sufficient to meet all the tests requirement. Therefore, the actual sample number (n) of each test item was marked in the test item table. Median and IQR range values are represented. All data can be found in Appendix A.

### 3.1. Volunteer Clinical Characteristics

Preterm infants’ clinical characteristics are shown in Table 1. Preterm infants in group A had lower gestational age (*p* < 0.0001) and birth weight (*p* < 0.0001) and were more critical. They were with higher incidence of NRDS (*p* = 0.034), application of mechanical ventilation (*p* = 0.001) and BPD (*p* < 0.0001), and longer stay in hospital (*p* < 0.0001). In group A, preterm infants had elder age (*p* = 0.011) and larger milk volume (*p* = 0.005), more preterm infants had reached EEN (*p* = 0.001) and lower incidence of sepsis (*p* = 0.046), but the weight (*p* = 0.004) was still lower. There was no significant difference in the application of prenatal antibiotics in mothers, the current application of antibiotics and probiotics in preterm infants, and the incidence of NEC between the two groups.

### 3.2. Original Fecal Characteristics

Original fecal characteristics are shown in Table 2. Group A had lower fecal pH, fecal ammonia, and fecal bile acids. Group B, meanwhile, had higher fecal sIgA level, both in OD and concentration methods.

### 3.3. Carbon Source Degradation Rate

Carbohydrates degradation rate in group A were higher than group B, especially in LAT medium (*p* = 0.017) and GOS medium (*p* = 0.005) (Figure 1). 

### 3.4. Gas Production

Gas production in different media is shown in Table 3 and Figure 2. Gas production amount, represented by air pressure difference, had no significant difference between the 2 groups in all 4 media.

### 3.5. SCFAs Analysis

Contents of several SCFAs in all media were different after fermentations (Figure 3). In LAT medium, total SCFAs (*p* = 0.011), acetic acid (*p* = 0.028), propionic acid (*p* < 0.0001), and butyric acid (*p* = 0.006) were higher in group A. In FOS medium, total SCFAs (*p* = 0.005), acetic acid (*p* = 0.005), and propionic acid (*p* = 0.003) were also higher in group A, while valeric acid (*p* = 0.037) was lower in group A. In GOS medium, valeric acid (*p* = 0.011) was also lower in group A. In GOS medium, propionic acid (*p* = 0.002) was higher in group A. In FL2 medium, butyric acid (*p* < 0.0001), isobutyric acid (*p* = 0.003), isovaleric acid (*p* < 0.0001), and total SCFAs (*p* = 0.039) were higher in group A.

## 4. Discussion

Breastfeeding is an ideal way to provide nutrition for healthy growth and development of infants, and is one of the important means to reduce the mortality of newborns, especially preterm infants. From February 2019, all the preterm infants less than 1500 g could be fed with human milk, which including mother’s own milk and DHM, even in the first few days, after we established our human milk bank. Many studies have shown that compared with formula milk, DHM can improve the feeding tolerance of infants, reduce the incidence of infection, NEC, BPD, and retinopathy of prematurity (ROP) [10] and reduce readmission and the risk of adolescent cardiovascular disease. Although pasteurization destroyed some components in DHM, the loss of nutrients was very small. Most of the non-specific immune substances were not affected. Therefore, DHM is a standard substitute for preterm infants when mother human milk is insufficient. 

It often takes 3 days before the mother can offer adequate human milk after an increase in lactation. Group A were preterm infants whose mothers had early lactation, as well as whose gestational age or weight was small enough to meet the acceptance criteria of human milk donation from the Human Milk Bank of Peking Union Medical College Hospital (PUMCH HMB, Beijing, China), and their parents agreed to accept human milk donation, and could receive human milk from initial feeding and throughout. Group B took preterm formula as the initial feeding method and changed to human milk after the mother’s lactation improved. So, in this research, the two groups of preterm infants only had differences in the initial feeding methods, but both were followed by human milk feeding. The diagnosis and treatment were in line with clinical routines, and there was no additional intervention. Therefore, the comparison can indicate the influence of initial feeding methods on the metabolites of intestinal flora in preterm infants.

Intestinal pH is the most representative index of intestinal environment; it plays a key role in bacterial activity and growth. In the distal colon, pH is higher than in the proximal colon, where SCFAs production may lower pH [11]. An in vitro study shows *Bacteroides* was inhibited at a pH lower than 6.5 and preferred alkaline pH for growth [12]. SIgA is the most abundant type of antibodies in intestinal secretions; it plays a key role in safeguarding the epithelium from invasive pathogens and commensal bacteria, resulting in a beneficial downregulation of inflammation [13]. In our study, preterm infants with human milk as initial feeding method had lower fecal pH. Like the situation in term infants [14], this indicates that human milk as initial feeding method can enhance the ability of preterm infants to resist pathogens by reducing intestinal pH. For sIgA, the initial methods did not play a decisive role. Preterm infants with more gestational weeks and more days after birth may be able to synthesize more sIgA due to better maturity of the immune system, which is consistent with the clinical differences in gestational age and age between group A and group B. Some intestinal metabolites maybe harmful and even be related to some diseases. As a metabolite of bacterial protein degradation and amino acid fermentation, fecal ammonia is toxic to intestinal tissue and causes inflammation in mammals [15,16]. Fecal bile acids, which are partially produced by intestinal microbiota, can promote the digestion and absorption of food. Bile acids are also signaling molecules. They are principally involved in metabolic regulation and energy metabolism [17], and they are linked to lipid and glucose metabolism via the farnesoid X receptor (FXR), a regulator of the enterohepatic circulation of bile acids [18]. Furthermore, fecal bile acids are considered potentially harmful, because some studies showed that they can stimulate oxidative stress and DNA damage due to their hydrophobicity [19] and induce the apoptosis resistance of colon epithelial cells [20], which could be related to both colorectal cancer etiology and inflammatory bowel diseases in adults [20,21,22,23,24,25]. Our data showed group A had a lower level of fecal ammonia and bile acids, which may indicate that initial feeding with human milk might lower the inflammation, oxidative stress, and DNA damage of intestinal or colon cells.

In the current study, the carbohydrates that mostly presented in human milk, including LAT, FOS, GOS, and 2-FL, were employed in the in vitro batch fermentation modeling system. Detecting the degradation rate of different carbon sources after batch culture fermentation could indicate the intestinal flora ability of utilization of these carbon sources. In our study, intestinal flora showed different rates of carbon source degradation based on different initial feeding methods. Group A had better carbon source utility in all four media and the differences were significant in GOS and LAT. FOS and GOS have become important infant food additives because of their low sweetness, low energy, difficult absorption, promoting the growth of intestinal probiotics, regulating intestinal microecology, and improving body immunity. GOS exists in breast milk, is not digested by human digestive enzymes, and has good thermal stability. Formula milk supplemented with GOS and FOS can help soften infant feces and stimulate the proliferation of beneficial bacteria such as Bifidobacteria and Lactobacilli in the intestine [26]. LAT is the predominant soluble digestible glycan in the milk. It provides a readily available energy source to newborn mammals and beneficial effects for gut physiology [27]. In young infants, LAT may reach the colon, where it is fermented to SCFAs which confer a range of beneficial prebiotic effects on the developing gut microbiome and intestinal barrier function [28]. Human milk oligosaccharides (HMOs) are the third most abundant solid component, after lactose and lipids, of human milk [29]. FL2 is the most abundant HMO secreted in human milk, accounting for about 30% of the total HMOs. HMOs are conducive to the normal digestion, absorption, secretion, and establishment of immune function in the intestine of infants [30,31]. At the same time, HMOs can also provide essential nutrients for brain development and cognition [32]. Thus, with the results of better carbon source utility in all four media in group A, especially in GOS and LAT, we deduce that intestinal flora of group A infants could make better use of these common oligosaccharides and might contribute more to the growth of beneficial bacteria, regulation of immune system, and development of other systems.

Compared with soluble metabolites, we pay less attention to gaseous molecules such as H_2_, CO_2_, NH_3_, CH_4_, NO, H_2_S, and CO. In fact, a large amount of gas produced by the human body every day is closely related to intestinal microbiome and their host [33,34,35,36]. For example, hydrogen, through the secondary metabolism of hydrogen-rich bacteria in the colon, can be processed into high concentrations of H_2_S, which can cause DNA damage to the host intestinal epithelial cells and destroy the intestinal mucosal barrier of the host [37]; the increase of methane content in the intestine and methane bacteria affects the intestinal motility, which is related to constipation symptoms [38]. Preterm infants are immature. Feeding intolerance, abdominal distention, and intestinal motility disorder are common problems in NICUs. It is a question whether gaseous molecules play a key role affecting the gastrointestinal function of preterm infants. Unfortunately, we did not find any difference in gas production amount between the two groups. On the one hand, initial feeding methods might not be the key factor for gas production. On the other hand, it might have influence on the component of gas, not the total amount.

In the meanwhile, contents of several kinds of SCFAs in all media were different after fermentation. In the large intestine, intestinal microbiota ferment indigestible dietary fiber and produce SCFAs, including acetic acid, propionic acid, butyric acid, isobutyric acid, valeric acid, and isovaleric acid [39]. SCFAs can be taken up by the gut epithelium and enter the portal circulation where they can be metabolized by the liver or released into the systemic circulation [40]. They have influence on the maintenance of gut-barrier function [41], blockage of the translocation of lipopolysaccharide [42], and gut and peripheral immune responses [43], inducing the differentiation of T-regulatory cells [44]. We can see group A had higher contents of most kinds of SCFAs in different media, which is also coincidental with the higher carbohydrate degradation rates. In contrast, valeric acid is lower in group A, especially in GOS medium. Since valeric acid can be produced by fermentation of proline, this result, together with the lower production of fecal ammonia, indicate that the microbiota in group A give preference to carbohydrate fermentation. Because SCFAs play various kinds of beneficial or harmful roles in their interactions with human body [45], this interesting result leads to further research on the influence of different individual SCFAs. 

Through the experimental results and correlation analysis, we can see the difference of intestinal microecology and microbial metabolites between different initial feeding methods group, even though they both continued with same feeding method, human milk. Group A had better carbohydrate utilization and higher SCFAs production, suggesting that human milk as initial feeding method can help preterm infants establish a better carbohydrate fermentation model. In contrast, Group B had higher ammonia production and valeric acid production, suggesting that formula milk as initial feeding method helped the intestinal flora of premature infants to establish a more visible protein fermentation model. We consider the reason to be the process of intestinal bacteria appearing and rapidly establishing the microbial community in the very beginning. Initial feeding methods will greatly affect the results of these processes through the formed intestinal environment in the first few days, which could be caused by the interaction of many factors, including intestinal pH or some intestinal metabolites such as fecal ammonia, bile acids, SCFAs, or even gas metabolites. Due to the existence of “the founder effect”, the differences of intestinal flora and metabolism caused by different initial feeding methods will exist for a long time, at least several weeks. Breastfeeding is an ideal way to provide nutrition for the healthy growth and development of infants. 

It is difficult to obtain more than 1 g of feces after absorbing water by diapers because preterm infants have light weight and less defecation. Although the sample size was small, the data were precious for us to explore the intestinal microbial metabolites of preterm infants. Therefore, the lower that the gestational age and age of preterm infant was, the less likely we were to be successful in obtaining feces or meeting the needs of all inspection items. This study was an observational study, so we did not take bowel lavage or other procedures to retain samples from preterm infants. Moreover, more ethical discussions will be needed if additional invasive operations are taken due to experimental requirements. For the evaluation of gas production, we only estimated the quantity. At present, with the development of technology, the identification and measurement for different kinds of gas can be carried out. Different gases have different impacts. If the gas kinds can be evaluated at the same time, the research will be more meaningful. Of course, intestinal metabolites are closely related to intestinal flora and other factors, such as maturity, feeding tolerance, infection, antibiotic use, probiotics, or probiotics use. In the future, we hope to collect more samples and data to explore the relationship between intestinal flora and microbial metabolites, and the experimental evidence level will be stronger.

## 5. Conclusions

The initial feeding methods may affect preterm infants’ ① intestinal microecology, including fecal pH, fecal ammonia, fecal bile acids, and carbohydrate utility; and ② microbial metabolites, for example, several different kinds of SCFAs in the long term.

## Figures and Tables

**Figure 1 microorganisms-10-01453-f001:**
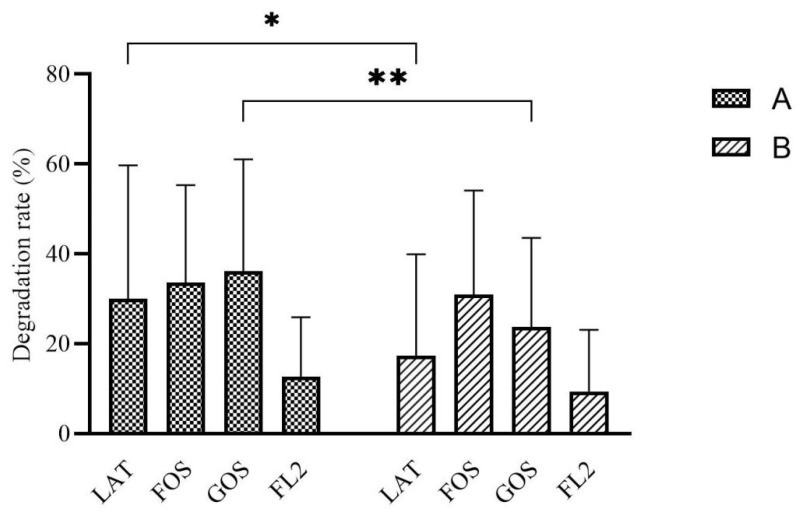
The rate of carbon source degradation by in vitro fermentation modeling system based on different initial feeding methods. Group A, preterm infants fed by human milk, including mother’s own milk and DHM; group B, preterm infants fed by preterm formula at first 3 days and fed by human milk (including mother’s own milk and DHM) from day 4 to discharge. LAT, lactose; FOS, fructooligosaccharide; GOS, galactooligosaccharide; FL2, 2′-fucosyllactose. Data are shown as median with interquartile and error bars represent 5–95 percentile (* *p* < 0.05, ** *p* < 0.01).

**Figure 2 microorganisms-10-01453-f002:**
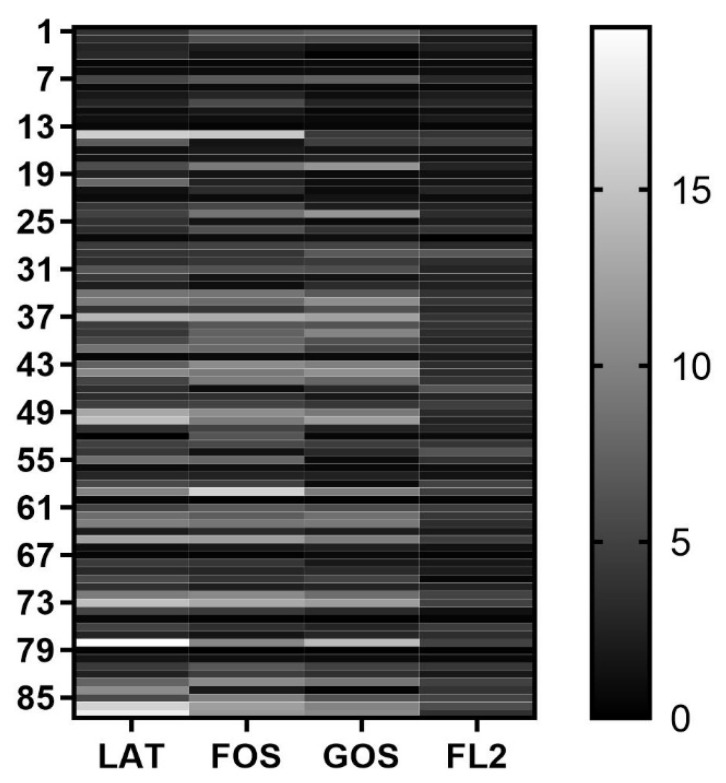
Heat map analysis of fecal samples based on gas production in different media by in vitro fermentation modeling system. Rows for different samples. Columns for different fermentation media. LAT, lactose; FOS, fructooligosaccharide; GOS, galactooligosaccharide; FL2, 2′-fucosyllactose.

**Figure 3 microorganisms-10-01453-f003:**
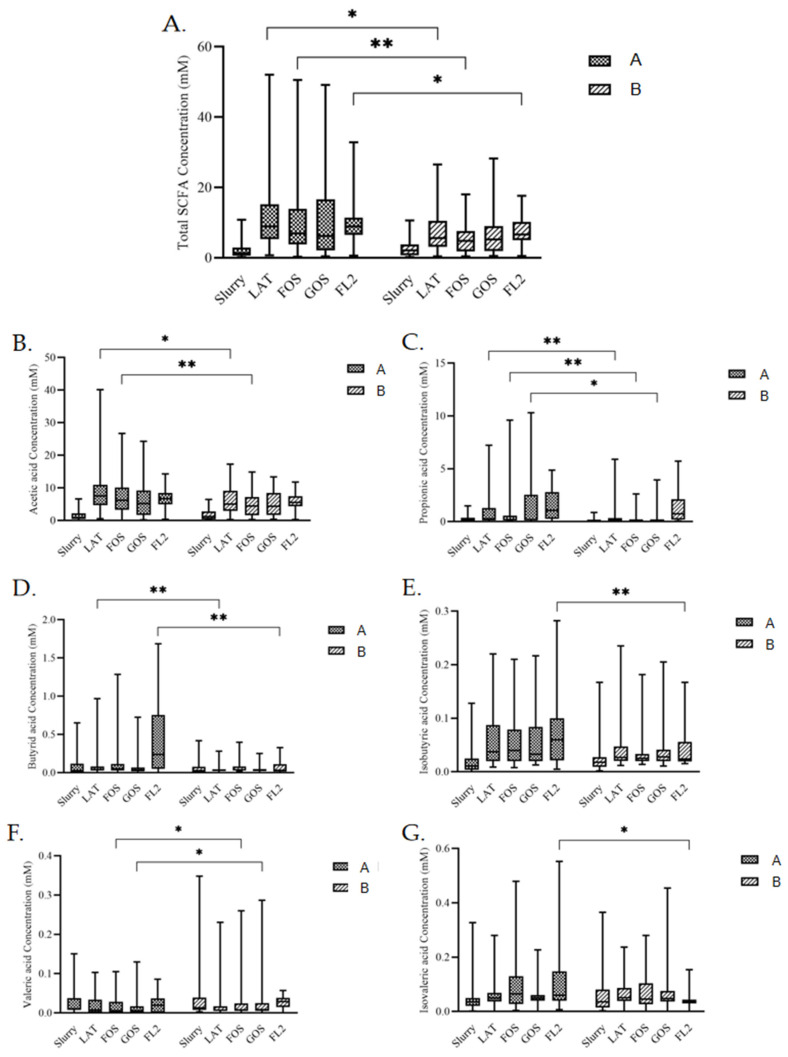
The concentration of fecal SCFAs, total SCFAs (**A**), acetic acid (**B**), propionic acid (**C**), butyric acid (**D**), isobutyric acid (**E**), valeric acid (**F**), and isovaleric acid (**G**) in different media by in vitro fermentation modeling system based on initial feeding methods. Group A, preterm infants fed by human milk, including mother’s own milk and DHM; group B, preterm infants fed by preterm formula for first 3 days and fed by human milk (including mother’s own milk and DHM) from day 4 to discharge. Slurry, 1:10 fresh fecal slurry; LAT, lactose; FOS, fructooligosaccharide; GOS, galactooligosaccharide; FL2, 2′-fucosyllactose. Data are shown as median with interquartile and error bars represent 5–95 percentile (* *p* < 0.05, ** *p* < 0.01).

**Table 1 microorganisms-10-01453-t001:** Preterm infants’ clinical characteristics.

Item	Results		
	Group A	Group B	Difference
n	90	70	
Gender			0.015
Male	62.2%	42.9%	
Female	37.8%	57.1%	
Gestational age (GA, weeks)	28.5 (25–29)	29.5 (28–31)	<0.0001
Birth weight (g)	950 (860–1173)	1255 (1005–1420)	<0.0001
Preterm premature rupture of membranes (PPROM)	37.8%	52.9%	0.057
Apgar score 1-Min			0.009
>7	82.2%	80%	
4–7	17.8%	20%	
Age (days)	28 (16–42)	22 (13–32)	0.011
Milk volume (ml/kg)	148 (95–164)	103 (65–151)	0.005
Exclusive Enteral Nutrition (EEN)	56.7%	28.6%	0.001
Weight (g)	1508 (1190–1831)	1748 (1491–2033)	0.004
Feeding intolerance	17.8%	14.3%	0.553
Using antibiotic	53.3%	64.3%	0.164
Using probiotics	38.9%	42.9%	0.612
History of sepsis	27.8%	42.9%	0.046
**Mother clinical characteristics**			
Mother age (years)	34 (31–36)	33 (32–38)	0.505
Diabetes			<0.0001
No	76.7%	48.6%	
Gestational diabetes mellitus (GDM)	23.3%	51.4%	
History of antibiotic in perinatal stage	96.7%	98.6%	0.444
**Diagnosis and treatment**			
Respiratory distress syndrome of newborn (NRDS)	94.4%	59%	0.034
History of invasive ventilator	60%	37.1%	0.001
Bronchopulmonary dysplasia (BPD)	67.8%	30%	<0.0001
Necrotizing enterocolitis (NEC)	10%	4.3%	0.173
Age reach EEN (days)	22 (16–25)	25 (18–33)	0.077
Discharge age (days)	64 (40–75)	42 (32–49)	<0.0001
Discharge weight (g)	2160 (2035–2545)	2120 (2020–2366)	0.115

**Table 2 microorganisms-10-01453-t002:** Original fecal characteristics.

Item	Initial Feeding Methods				Difference
	Group A		Group B		
	n	Value	n	Value	
Fecal pH	80	6.24 (5.62–6.52)	45	6.37 (5.83–6.89)	0.023
SigA (OD)	82	0.375 (0.286–0.752)	38	0.670 (0.340–1.108)	0.046
SigA (concentration)	82	0.000110 (0.00000000218–0.000569)	38	0.00132 (0.00045–0.00243)	<0.0001
Fecal ammonia (μmol/g)	82	4.31 (2.28–8.76)	38	9.08 (4.45–13.57)	0.001
Fecal bile acids (μmol/g)	82	0.34 (0.31–0.38)	38	0.36 (0.34–0.40)	0.025

Group A, preterm infants fed by human milk, including mother’s own milk and DHM; group B, preterm infants fed by preterm formula for first 3 days and fed by human milk (including mother’s own milk and DHM) from day 4 to discharge; n, sample number.

**Table 3 microorganisms-10-01453-t003:** Gas production in different media by in vitro fermentation modeling system.

Air Pressure Difference (kpa)	Initial Feeding Methods				Difference
	Group A		Group B		
	n	Value	n	Value	
LAT	89	3.8 (1.25–5.75)	70	4.0 (1.2–6.3)	0.812
FOS	89	4.10 (1.65–7.35)	70	3.3 (1.2–7.5)	0.207
GOS	89	2.80 (0.95–6.20)	70	1.9 (0.7–7.0)	0.378
FL2	80	2.85 (1.53–4.00)	45	2.8 (2.2–4.0)	0.934

Group A, preterm infants fed by human milk, including mother’s own milk and DHM; group B, preterm infants fed by preterm formula at first 3 days and fed by human milk (including mother’s own milk and DHM) from day 4 to discharge; n, sample number; LAT, lactose; FOS, fructooligosaccharide; GOS, galactooligosaccharide; FL2, 2′-fucosyllactose.

## Data Availability

Publicly available datasets were analyzed in this study. This data can be found in the Appendix A.

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
