# Peer review of "Evaluation of Intestinal Microbial Metabolites in Preterm Infants with Different Initial Feeding Methods by In Vitro Fermentation Modeling System"

_microorganisms, 2022, doi:10.3390/microorganisms10071453_

Round 1
Reviewer 1 Report
The colonization of infant gut microbiota has been widely recognized as akey factor in brain development and shaping the immune system. The composition
and development of infant gut microbiota can be influenced by many prenatal factors,
such as maternal diet, obesity, smoking status, and use of antibiotic agents during
pregnancy and mode of delivery. Infants born vaginally have a gut microbiome
very similar to that of their mother's vaginal and faecal flora. This occurs through
vertical transfer of the vaginal-perianal microbes of the mother as the infant passes
through the birth canal. The role of preterm microbiota and first it establish in
the first day of life is constantly being researched. The article presented for
review described very interesting and relevant aspects of neonatal microbiota
research. It is worth emphasizing to make other researchers aware of how valuable
a diagnostic source of microbiota is feces and how many parameters can be tested
from this material in order to assess the microbiota. There is prospective observational
study. In preterm newborn with gestational age lower than 34 week of gestational
age group A, preterm infants fed by human milk, including mother’s own milk
or donor milk ; group B, preterm infants fed by formula at first 3 days, including
preterm formula and post discharge-formula, and fed by human milk
(including mother’s own milk and DHM) from Day 4 to discharge.
Non-invasive methods of the faecal samples were collected every 7 days
until the infants were 34 corrected gestational weeks or discharged. Material
160 faecal samples (group A included 90 samples and group B) included
70 samples( were representative. The authors analyzed a very significant
number of parameters. This is extremely rare in neonatal literature and very
valuable. The authors made the measurements; faecal pH SIgA(concentration,)
ammonia, bile acids. LAT ,FOS GOS ,FL2. Assessment methods included;
Batch culture fermentation, thin-layer chromatography, Short chain fatty acids analysis,
(ELISA).This indicates a very rich methodological workshop that may be a hint
for other researchers. This interesting result leads to further research on the influence
of different individual SCFAs. The authors included 3 figures and 5 tables in the work
that present the results well. The discussion was conducted in a very interesting way.
Very rich literature containing 42 items. The items are well-chosen and current. For
all above reasons, this work is very valuable. However, the manuscript has some clinical weaknesses. 1) Authors should be taken into account that the nature of maternal microbiota
during pregnancy can influence initial neonates colonization and may contribute
to dysbiosis leading to disease. Ideal conditions for initial colonization results
in a symbiotic relationship between the colonizing bacteria and intestinal epithelial
and lymphoid tissues and immune and metabolic homeostasis during pregnancy.
Because of that authors should include to study examination of the first meconium. 2) Clinical material -the groups of newborns- are very heterogeneous and this can
affect the final results. There are many differences in perinatal and neonatal
factors between the two groups of newborns. For example: in group B was more
incidence of preterm rupture of membranes (PPROM) 37.8% versus 52.9% .
Group B received less volume of the milk and has less exclusive enteral nutrition
(EEN) 56.7% versus 28.6% 3) Why authors did not use used propensity score matching (PSM) to adjust for
the impact of baseline characteristic differences between the groups. 4) Line 98,99 “group B, preterm infants fed by formula at first 3 days, including
preterm formula and post-discharge-formula, and fed by human milk (including
mother’s own milk and DHM) from Day 4 to discharge”. The authors should
explain and specify why newborns from group B during the first 3 days of life
received not only the preterm formula but also the post discharge formula. This
is completely incomprehensible. 5) The results would be more reliable if the authors limited clinical material to
newborns <28 weeks gestation. Both groups were fed maternal milk or milk
from donors until the day of discharge or until 34th corrected week, beyond
the first 3 days of the children from group B. Because the feces were examined
every 7 days up to the 34 corrected gestational weeks or discharged the newborns
born in newborns born in 33 weeks only one stool sample was collected.
6) At the end of the paper does not have a clear conclusions. In abstract authors
wrote: “The initial feeding methods may affect the intestinal microecology
microbial metabolites in the long term and further affect the feeding tolerance,
establishment of intestinal barrier, and even the health of many systems” .
It looks like conclusions but it do not match the results This does not follow
from the work. Authors did not provide a follow-up and do not examine
the long term affect the feeding tolerance.
Reviewer 2 Report
Title. Maybe its better, by in vitro fermentation modeling system
L 49-51: “The metabolites of human intestinal microorganisms include not only various soluble metabolites, but also different types of gas [3], which can act on the digestive system and even other human systems at the metabolic level.” The production of gases is not considered metabolite but a by-product of metabolism. I cannot understand what you mean maybe, higher levels of propionic acid and butyric acid production.
L 95: “ ..were transferred to other hospitals or died during hospitalization”. Since you are talking about newborns, I think it is wiser to use the expression passed away, instead of died.
L 98-99: «Group B, preterm infants fed by formula at first 3 days, including preterm formula and post-discharge-formula». What is the composition of the administered formula?
L 111: “..and fecal pH was tested directly by pH meter.” By which method can we measure pH in samples, where the material is in solid form. I think it's wrong. Please correct it.
L 129: “Rest fresh fecal slurry”. The remaining fresh fecal slurry
Section Batch culture fermentation
L 117-130: The cited reference, refers to adults. Have you made any modifications to the use of these growth mediums to study the newborn intestine tract? Because microflora of newborns and adults are two different things.
Please describe in detail the methodological approach of the section Batch culture fermentation, not just the using mediums.
Where is the enumeration of the bacterial populations? Without recording the gut microbiota, I think there is a big gap in the methodology.
Table 2: What is the n?
Table 3: There should be an interpretation text of all the data placed in the table.
Table 4: Title; Gas production in different using media, by in vitro fermentation modeling system; There should be an interpretation text of all the data placed in the table.
Table 5: “Table 5. Different SCFAs in different media”. Please revise the title. Also, what is the n?
L 294: “Compared with soluble metabolites, we pay less attention to gas metabolites usually”. I know that there are metabolic products produced by the gut microbiota, where during their production we have gas emissions. So, gas metabolites, I don’t think so.
Discussion: I think the discussion is incomplete. Without knowledge of the existing microbiota of each group and their nutritional differentiation, cannot be a substantive study of the results.
Where are the section Conclusions?
Round 2
Reviewer 2 Report
I firmly believe that the production of gaseous molecules cannot be considered metabolite, but a by-product of metabolism. We have secondary metabolites which are natural products synthesized mainly by bacteria, fungi, and plants and their production is not necessary for the growth and reproduction of organisms, but play key roles in the survival of the organisms that produce them, because they determine interactions within their environment. The gaseous molecules produced are end-product of a metabolic pathway without any benefit to the body, destined to be released into the environment.
L. 221: n (not N); same L 223; L 224; L 262
Author Response
- I firmly believe that the production of gaseous molecules cannot be considered metabolite, but a by-product of metabolism. We have secondary metabolites which are natural products synthesized mainly by bacteria, fungi, and plants and their production is not necessary for the growth and reproduction of organisms, but play key roles in the survival of the organisms that produce them, because they determine interactions within their environment. The gaseous molecules produced are end-product of a metabolic pathway without any benefit to the body, destined to be released into the environment.
Reply:Thank you for your detailed explanation. I have learned a lot and made changes according to it.
- 221: n (not N); same L 223; L 224; L 262
Reply: Thank you for your reminder, I corrected them all.